# Dlg Is Required for Short-Term Memory and Interacts with NMDAR in the *Drosophila* Brain

**DOI:** 10.3390/ijms23169187

**Published:** 2022-08-16

**Authors:** Francisca Bertin, Guillermo Moya-Alvarado, Eduardo Quiroz-Manríquez, Andrés Ibacache, Andrés Köhler-Solis, Carlos Oliva, Jimena Sierralta

**Affiliations:** 1Department of Neuroscience and Biomedical Neuroscience Institute, Faculty of Medicine, Universidad de Chile, Santiago 8380453, Chile; 2Department of Cellular and Molecular Biology, Faculty of Biological Sciences, Pontificia Universidad Católica de Chile, Santiago 8331150, Chile

**Keywords:** synapse, *Drosophila*, short-term memory, post-tetanic potentiation, electrophysiological, post-tetanic depression, DlgS97, DlgA, neuromuscular junction, NMDAR

## Abstract

The vertebrates’ scaffold proteins of the Dlg-MAGUK family are involved in the recruitment, clustering, and anchoring of glutamate receptors to the postsynaptic density, particularly the NMDA subtype glutamate-receptors (NRs), necessary for long-term memory and LTP. In *Drosophila*, the only gene of the subfamily generates two main products, *dlgA*, broadly expressed, and *dlgS97*, restricted to the nervous system. In the *Drosophila* brain, NRs are expressed in the adult brain and are involved in memory, however, the role of Dlg in these processes and its relationship with NRs has been scarcely explored. Here, we show that the *dlg* mutants display defects in short-term memory in the olfactory associative-learning paradigm. These defects are dependent on the presence of DlgS97 in the Mushroom Body (MB) synapses. Moreover, Dlg is immunoprecipitated with NRs in the adult brain. Dlg is also expressed in the larval neuromuscular junction (NMJ) pre and post-synaptically and is important for development and synaptic function, however, NR is absent in this synapse. Despite that, we found changes in the short-term plasticity paradigms in *dlg* mutant larval NMJ. Together our results show that larval NMJ and the adult brain relies on Dlg for short-term memory/plasticity, but the mechanisms differ in the two types of synapses.

## 1. Introduction

Synaptic plasticity in general, and more specifically the regulation of intracellular calcium, is believed to be the cellular mechanism forming new memories in the brain [1,2]. Even more, the molecular mechanisms that allow the diverse forms of synaptic plasticity and the formation of memories are shared through evolution [3,4]. In *Drosophila*, short-term memories can be formed by a conditioning paradigm, based on the pairing of aversive stimuli (unconditioned stimuli) and an odor (conditioned stimuli). This process depends on the activity of the mushroom body structure of the central brain, in which the convergence of sensory inputs and dopaminergic modulatory inputs [5,6,7] allows the generation of input-specific synaptic modifications.

The short-term synaptic plasticity mechanisms rely on the presynaptic regulation of the neurotransmitter release, which in turn is determined by the calcium concentration inside the terminal [8,9]. Thus, the calcium dynamic is controlled by the entering of the calcium ions through the voltage-gated calcium channels (VGCC) in the plasma membrane, and by the sequestration of the calcium in internal stores, the calcium buffering by cytosolic proteins, and its extrusion to the extracellular space through ATP-dependent transporters regulates the short-term synaptic plasticity [10,11]. On the other hand, the long-term synaptic plasticity is regulated by the postsynaptic mechanisms, including the calcium entrance through the NMDA-type glutamate receptors (NRs), as well as the recruitment of the receptors and regulators of their activity. At the pre- and postsynaptic compartments, the plasma membrane channels, transporters, and pumps are assembled in place by the scaffolding proteins. The *Drosophila* Discs Large (Dlg) is a scaffolding protein that regulates the size of the active zones and the post-synaptic receptor’s fields, and has been associated with calcium regulation [12,13].

In *Drosophila*, the Dlg-MAGUK family of scaffolding proteins is essential for the formation of the multiprotein complexes at the cell–cell interaction domains and plays an important role in synaptic physiology [12,13,14,15]. The in vivo role of these proteins in vertebrates, although extensively studied in vitro, is not yet completely clear, in part, due to the existence of four genes in vertebrates that show a high degree of redundancy [16]. The use of *Drosophila melanogaster* overcomes this difficulty, since it possesses only one gene for the Dlg-MAGUK family, *dlg1* (*dlg*). In *Drosophila* and vertebrates, the main *dlg* products are proteins with or without an S97 domain, a protein domain that allows the formation of a heterocomplex with other proteins with the same domain. *Drosophila*’s main *dlg* products are DlgS97 and DlgA [13], with or without the S97 domain, respectively. They are both expressed at the *Drosophila* larval neuromuscular junction (NMJ), a glutamatergic synapse that shares all of the fundamental molecular components with the vertebrate’s central nervous system synapses [17,18]. The *dlg*-mutant flies display several phenotypes, depending on the missing protein. While the *Drosophila dlgA* null mutant flies (*dlgA*) are late larval lethal, the *dlgS97* mutant flies are viable but display several altered behaviors, including the disruption of circadian rhythm, decreased courtship, and decreased geotaxis [15,19,20]. Additionally, the synaptic transmission at the larval NMJ is altered in *dlg* mutants, exhibiting defects in the short-term plasticity paradigms, such as increased pair pulse facilitation and augmentation during tetanus in low extracellular calcium conditions [13]. However, under high synaptic vesicle fusion probability closer to physiological conditions (1.5 to 2 mM extracellular calcium concentration), the basal-evoked release is not different from the control [13]. In these conditions, the synaptic plasticity becomes more complex, where facilitation, depression, and augmentation all play a role in synaptic plasticity [21,22].

In *Drosophila*, the requirement of the scaffolding proteins in the processes of learning and memory, and their association with short-term plasticity, has been scarcely explored; however, considering the structural and functional synaptic alterations observed in the *dlg* mutants, it is possible to suggest that these synaptic modifications could also associate with defects in these complex processes. Even more, taking all of the above into account, we hypothesized that the *Drosophila* animals lacking one or both isoforms of *dlg* have defects in memory formation and synaptic plasticity under physiological conditions.

Here, we show that the *dlgS97* mutants and knockdown display defects in short-term memory in the classic olfactory associative, aversive and appetitive, learning paradigm for adult animals [5]. These defects are dependent on the presence of DlgS97 in the mushroom bodies’ synapses, but do not rely on the presence of this protein in the ascending pathway. Moreover, we determined that the Dlg immunoprecipitated with the NRs in the adult brain and that the localization of the NR-tails constructs, retaining its PDZ motifs in the synapse, depends on Dlg. Thus, the NRs associate with the Dlg in the adult brain, suggesting that the short-term memory defects are a consequence of the disruption of this association. On the other hand, the *dlg* mutants, display changes in the plasticity paradigms in the larval NMJ under conditions of a high probability of vesicle fusion. We determined that the NRs in the NMJ are not present, supporting a role for the Dlg in short-term memory independent of the NRs. Together, our results show a strong dependence on Dlg for short-term memory in adult flies, and suggest that the way in which Dlg plays its role could be independent or associated with NRs.

## 2. Results

### 2.1. Dlg Modulates the Time Course of Augmentation and Depression

The classic short-term synaptic plasticity paradigms, such as pair pulse and facilitation, depend on the presynaptic residual calcium and persist for no more than a few seconds [23,24]. The other forms of short-term plasticity, such as post-tetanic potentiation (PTP) and augmentation, are two closely related forms of enhancements, elicited by high-frequency stimulation. While PTP could last for 30 s to minutes, the augmentation lasts for 1 to 10 s [22]. PTP is thought to depend on calcium-regulated changes and includes an increase in the number of vesicles ready to be released (RRP), or in the probability of release [23]. On the other hand, depression is another form of synaptic plasticity, observed as a decrease in synaptic efficiency, which occurs during the tetanic stimuli, depending on the level of activation obtained during the conditioning episode. During a tetanic stimulus, the total synaptic plasticity is the result of the sum effects of augmentation, PTP, and depression. PTP is easily observed under low-release probability conditions (low extracellular calcium concentration, i.e., 0.2 mM), but at a higher release probability (more than 1 mM extracellular calcium concentration), the responses to the sustained stimulus show an increased depression, resulting in an overall decreased neurotransmitter release (post-tetanic depression, PTD), although augmentation is still present. Actually, under this condition, the augmentation is a major factor acting to sustain the neurotransmitter release in the presence of depression [21]. At 2 mM external calcium concentration, a concentration close to the physiological concentration [25], the response to tetanus is observed as a fast depression that tends to recover after the end of the stimulation, but does not reaches pre-tetanic amplitudes, resulting in a PTD.

Previously, we have shown that the short-term plasticity observed in the paired-pulse was affected in the *dlg* mutants, i.e., at 0.2 mM external calcium concentration, there was increased facilitation and at 2 mM external calcium concentration, there was an increased depression [13]. To assess whether the *dlg* mutants display a difference in the post-tetanic responses under high neurotransmitter release probability (2 mM Ca^2+^ concentration in the bath), we used a stimulation paradigm consisting of 25 stimuli at 0.5 Hz to establish the basal amplitude response, followed by 1000 stimuli at 20 Hz for conditioning, and ending with 100 stimuli at 0.5 Hz to evaluate the post-tetanic plasticity (Figure 1A,B). The amplitudes were normalized to the average pre-tetanic response (Figure 1C). To quantify the tetanic plasticity, we compared the normalized amplitudes of the first response after the tetanic conditioning (Figure 1B,C). We compared the control strain with the splice form-specific mutant *dlgS97*, *dlgA*, [15] and the hypomorphic strain *dlg^XI-2^* [14]. The initial value after the tetanus was in folds 0.47 ± 0.077 in the control; 0.39 ± 0.077 in *dlgS97^5^*; 0.56 ± 0.089 in *dlgA^40.2^*; 0.50 ± 0.119 in *dlg^XI-2^*, showing no statistical differences between the mutants and the control (Figure 1D). Additionally, we determined the augmentation quantifying the highest normalized value reached after the conditioning tetanus (Figure 1E,F). The average augmentation value was in folds 0.78 ± 0.1, in the control; 0.70 ± 0.02, in *dlgS97*; 0.80 ± 0.08, in *dlgA*; 0.85 ± 0.08, in *dlg^XI-2^*, again, no significant difference between the mutants and the control was found. Finally, we quantified the PTD by averaging the last ten responses after the tetanic stimulation (average stimuli between numbers 1116 to 1125). The final value was in folds 0.63 ± 0.11, in the control; 0.54 ± 0.07, in *dlgS97*; 0.66 ± 0.07, in *dlgA*; 0.64 ± 0.06, in *dlg^XI-2^* (Figure 1F) Neither of these values presented statistical differences between the *dlg* mutants and the control (Figure 1C–F), showing that the *dlg* mutants display post-tetanic plasticity in the same way as the control animals.

Another form of synaptic plasticity is post-tetanic plasticity, which includes the augmentation as well as depression components. We hypothesized that these post-tetanic response components could be affected in the *dlg* mutants. Thus, to reveal both of the components, we used a previously described model that incorporated depression in a multiplicative function Augmentation (A)*Depression (D), which generates two exponential components, in such a way that the post-tetanic plasticity is defined as PTP = (1 + A × e^−t/τA)^) × (1 − D × e^−t/τD^); where τA is the augmentation and τD is the depression time constant [21]. To fit the post-tetanic responses (Figure 1G) to this function, we used a non-linear fit corresponding to the sum of the two exponential functions (red line in Figure 1G and blue line in Figure 1H). To aid this nonlinear fit, we incorporated the values of the depression obtained during tetanic stimulation. With this model, we obtained the values for the time constants of decay of the augmentation, and the recovery from depression. The representative examples of the non-linear fit for augmentation and post-tetanic depression are shown in Figure 1H. For augmentation (grey line in Figure 1H), all of the *dlg* mutants show time constants that were significantly slower than the control (Figure 1I); in other words, that the time to recover the normal amplitudes after reaching an augmented response is longer. The average time constant (τA) was (mean ± SD) 32.3 ± 3 seg for the control; 49.9 ± 4.8 seg for *dlgS97*; 49.3 ± 11.4 seg for *dlgA*; 40 ± 13 seg for *dlg^XI-2^*. In contrast, the time to recover from the depression (red line in Figure 1H) values of the *dlgS97* and *dlgA* mutants was significantly faster compared with the control (Figure 1J). The average time constant (τD) was 12.8 ± 1.6 seg for the control; 6.9 ± 2.1 seg for *dlgS97*; 8.3 ± 0.96 seg for *dlgA*; 7.3 ± 1.97 seg for *dlg^XI-2^*. These results show first, that all of the *dlg* mutants can sustain post-tetanic plasticity and second, that the *dlg* mutants exhibited a faster recovery from the tetanic depression and a slower decay from the tetanic augmentation.

### 2.2. Dlg Mutants Show Altered Short-Term Memory

Considering the previous results using low probability release conditions [13], we hypothesized that the behavioral features that require this type of plasticity, such as short-term memory, should be altered. However, our results using high probability release conditions only supported mild defects in the short-term plasticity. To investigate whether the short-term memory was affected, we started by determining whether the adult *dlgS97^5^* mutants displayed structural defects in the brain. As seen in Figure 2A–F, our gross morphological analysis did not find significant defects in the mushroom body neuropil or the glial or neuronal structure or distribution.

To determine if a lack of Dlg does associate with defects in the short-term memory, we tested a Pavlovian conditioning paradigm in the *dlgS97* mutants, the only adult *dlg* mutant viable, and the main *dlg* variant expressed in the nervous system, which displays a reduction of more than 90% in the Dlg expression in the central brain [15]. We performed the well-known aversive learning paradigm [5] in young adult flies, in which an electric shock is paired with an odor (Figure 2(G1)). The control flies displayed a robust avoidance of the conditioned odor, while the *dlgS97* mutants showed a strong defect in the performance in this paradigm, with a Learning Index (LI) significantly lower than the controls (Figure 2(G2)). To determine if the second type of memory is also affected, we performed the appetitive paradigm, in which starved flies are presented with an odor and food at the same time. The control flies displayed a strong attraction to the food-paired odor, however, the *dlgS97* mutants were not attracted to the food-associated odor, showing that, despite the short-term memory paradigm performed, the *dlg* mutants are unable to form this type of memory (Figure 2(G3)).

In *Drosophila*, the associative learning is highly dependent on the function of the Kenyon neurons, which form the mushroom bodies neuropil in the adult brain [26]. To confirm the specificity of the function of Dlg in the Kenyon cells, we expressed a UAS-*dlg*-dsRNA (*dlg*-RNAi) construct, which targets both of the Dlg isoforms, in the olfactory receptors, projection neurons, and all or the subgroups of the Kenyon cells (Figure 2H,I), all of the structures with high DlgS97 expression. First, we expressed *dlg*-RNAi in all of the neurons (elav-GAL4) and were able to replicate the defects observed in the mutants. Then, we used two different drivers to evaluate the effects of the *dlg*-RNAi expression in all of the Kenyon cells, 117Y-GAL4 driver, and OK107-GAL4, which enables a broader expression, including the Kenyon cells, the projection neurons and the interneurons of the antennal lobe (Figure 2I). We observed that the expression of *dlg*-dsRNA with 117y-GAL4 driver decreased the significantly aversive olfactory learning (Figure 2I), while with OK-107, the tendency was not statistically significant. Although this is puzzling, since OK107 is the driver that includes more of the cells in the MB, the significance could have been affected by the dispersion in this strain. In addition, the strength of the expression of the RNAi construct could be better in the 117y line. Interestingly, the expression of *dlg*-RNAi in the subsets α′/β′ of the mushroom bodies using the C305A-Gal4 driver does not disrupt the associative learning, suggesting that the learning defects only occur when knockdown is performed in the whole structure. To confirm these results, we performed rescue experiments, expressing DlgS97 in the mushroom body, using the OK107-GAL4 driver in the *dlgS97* mutant background (Figure 2J). In this experiment, since *dlg* is in the X chromosome, and these rescue experiments required the development of a strain also carrying the GAL4-UAS construct, we only used male flies. Our results show that the mutant animals expressing DlgS97 in the Kenyon Cells do not show significant defects in short-term memory, supporting a role of Dlg in the MB for short term memory. These memory defects do not rely on the presence of Dlg in the ascending pathway, as we did not observe a decrease in performance by the expression of the *dlg*-dsRNA construct in all of the olfactory neurons using the Orco-GAL4 driver, or by its expression in the projection neurons using the drivers GH146-GAL4 and Tenm-GAL4 (Figure 2H). These results support an essential role of DlgS97 in the Kenyon cells to generate short-term associative memory.

### 2.3. NMDA Receptors Associate with Dlg

It has been reported that the vertebrates and invertebrates’ glutamate receptors, particularly of the NMDA subtypes (NRs), are essential for the establishment of memory [27,28,29]. Additionally, in mammals, PSD95 (hDLG4), one of the four ortholog genes to Dlg, binds directly to the NRs. In the *Drosophila* adult heads, the NRs co-precipitated with Dlg [30]. We confirmed these results by performing an immunoprecipitation, using anti-Dlg or anti-NMDAR2 [31] antibodies, and looked for NR or Dlg in the precipitated complex. Figure 3A shows that, regardless of the antibody used to precipitate the protein complex, both of the proteins precipitate together in the adult heads (Figure 3B).

To explore the type of association between the two proteins, we looked for the presence of NR in larval NMJ, where our group and others have reported different defects in the *dlg* mutants [19,32]. However, we were not able to detect the presence of the receptors R1 or R2 using the antibodies that can detect the proteins in WB and immunolocalization in the adult brain. To make sure that no NRs were present, we stimulated the NMJ preparation with NMDA in the absence of magnesium and recorded the calcium signals in the muscle. We detected strong responses in the muscle to the NMDA when the ventral cord was attached, however, the NMDA in the media did not elicit any response when the brain and ventral nerve cord (VNC) were removed (Appendix A). It is important to highlight that the remotion of the VNC does not perturb the structure or function of the NMJ. Additionally, the area stimulated is close to the VNC, since we use the A2 segment of the larvae—the more cephalic segment. Therefore, these results, in our view, confirmed the absence of the receptors to NMDA in the muscle, and suggest a modulatory function of NRs in the motoneurons in the ventral cord.

The *Drosophila* NMDAR display a PDZ motif in their carboxy terminal tail, which suggests that, as with the vertebrates, these receptors are able to bind to the DLG-MAGUKs. Therefore, taking advantage of the absence of NRs in the muscle and using the C57-Gal4 muscle-specific driver, we expressed in this tissue the tails of the dNMDAR2 and dNMDAR1 fused to CD8-GFP, to determine if their synaptic localization depended on Dlg (a similar type of experiment was performed in [33]). To determine the enrichment in the subsynaptic reticulum (SSR), we obtained a ratio between the fluorescence in the synaptic compartment and the fluorescence in the muscle. As is observed in Figure 3C,G, a control construct CD8-GFP shows the fluorescence in the membrane-rich SSR, and a similar fluorescence signal was observed for the dNR1-tail expressed in the muscle (Figure 3D,G). In turn, the dNR2-tail expressed in the muscle shows an enrichment in the synapse (Figure 3E,G). Importantly, when both of the tails are co-expressed in this tissue, the enrichment in the synapse is greater than that observed for the dNR2-tail by itself (Figure 3F,G). These results suggest that the dNR1 and dNR2 carboxy-terminal tails form a complex to be destined for the synapse, where the dNR1 tail is retained in the internal membranes and the dNR2 is destined for the synapse.

To determine the dependence on Dlg of the synaptic enrichment, we expressed the dNRs tails in WT, as the control, and the null mutant animals for *dlgA*, *dlgS97*, and the strong hypomorph *dlg^XI-2^* (Figure 4). As shown in Figure 4A,C,E,G, the *dlgA* mutants do not seem to affect the localization of the dNR1 or dNR2 tails, since their distribution is similar in the control and mutant animals. However, our results displayed in Figure 4B,F show that in the *dlgS97* mutants, the dNR1 is more enriched in the synapse and the dNR2 is less enriched compared to the control WT. Figure 4I displays the quantification of the fluorescence intensity as a ratio of the fluorescence immediately next to the HRP label (peri-synaptic compartment) and the fluorescence 3 µm from this signal. A number close to one indicates that the signal is homogeneous around the bouton and through the SSR, a number lower than one indicates that the signal is more abundant away from the bouton (3 µm), while a higher number would signify that the signal is very concentrated next to the bouton. The Figure shows that the overexpression of the dNR1 in the WT or *dlg* mutants does not change the distribution in the SSR, while in the case of dNR2 the results show that its distribution is affected in the *dlgS97* mutants and *dlg^XI^*^-2^, but not in the *dlgA* mutants, indicating a reduced fluorescence signal in the peri-synaptic compartment of the bouton; thus, supporting a role for DlgS97 in allowing dNR2 to reach the synapse, especially the region close to the bouton. On the other hand, Figure 4J quantitates the ratio between the SSR and the muscle, thus showing the enrichment of the signal over the distribution in the internal membranes. A lower ratio compared to the WT control indicates that the signal in the internal membranes increased with respect to the synaptic signal. Figure 4J shows that the hypomorph mutant presents a decreased ratio for dNR1, which means that in this mutant, the dNR1 is more retained in the internal membranes, while for dNR2, only in the *dlgA* mutants do we see the same phenotype. These experiments support a role for *dlgS97* in the dNR2 destination to the synapse, and dlgA could have a role in allowing the exit of dNR2 from the ER. Together, these experiments give support to the association of DlgS97 to the PDZ-motif-containing dNR2 tail and suggest a direct relationship between these two proteins. Additionally, adding the immunoprecipitation assays shown here, and those reported previously, together with defects in short term memory observed in the dNRs mutants, it offers a possible mechanism by which the Dlg could be affecting the short-term memory in adult flies.

## 3. Discussion

### 3.1. Short-Term Plasticity at the NMJ

The communication between the neurons is key for memory and learning processes. The modulation of the neurotransmitter release properties enables plasticity and broadens the diversity of the neural circuits to respond to changing inputs. The modifications in the dynamics of Ca^2+^ in the presynaptic cells changes the synaptic homeostasis [34,35] and plasticity [22,23,36]. Dlg has been linked with changes in the Ca^2+^ channel localization, along with changes in the Ca^2+^ dependent quantal content and plasticity under low neurotransmitter-release probability conditions [13]. However, here we show that under high neurotransmitter release probability, closer-to-physiological conditions [25], the *dlg* mutants can compensate for the synaptic defect in calcium entrance and do not change the overall postsynaptic response during a plasticity-inducing stimuli paradigm (Figure 1A–D). The plasticity protocol used in the experiments at 2 mM extracellular calcium increases the bulk of Ca^2+^ inside the terminal during the tetanic stimulation, inducing depression during the tetanus and post-tetanic depression [23]. Although we did not observe modifications in the amplitude of the post-tetanic plasticity parameter, changes in the time constant for the post-tetanic augmentation and depression (Figure 1I–J) were observed, suggesting that the normal plasticity observed is the result of complementary compensations due to an altered Ca^2+^ dynamic, with a consequence in the time-course of the plasticity. The time constant for the decay of augmentation has a strong correlation with the changes in the dynamics of the bulk Ca^2+^ inside the presynaptic terminal during repetitive stimuli, which in turn affects the probability of neurotransmitter release [23]. The calcium accumulated during the tetanus increases the probability of release; upon the termination of the tetanus, the probability of release will decrease following the kinetics of the return to basal calcium, which is thought to determine the augmentation time constant. The slower time constant thus reflects a slower return to the basal release probability, which in turn reflects a slower return to the basal calcium concentration. These slower kinetics for the calcium were previously observed in the *dlgS97* mutants using calcium indicators [13], and could be related to a slower extrusion of calcium or deficient calcium buffering mechanisms. On the other hand, the depression reflects the depletion of the RRP and how fast the system is capable of replenishing this pool, a mechanism that is also dependent on the level of Ca^2+^ inside the terminal [21,22,37]. Our data show that the depression recovery time is faster in the two variant-specific mutants compared with the control animals. This supports that the decreased RRP takes less time for the replenishment of the vesicles in the *dlg* mutants.

Taken together, the data suggest that the *dlg* mutants showed an increase in the Ca^2+^ concentration during the tetanic stimulation, which is greater than the control flies. The changes in short-term plasticity, such as PTP and Ca^2+^ homeostasis, were linked to neurodegeneration [38] and impairments of memory and learning [39,40] in mammals. Additionally, the calcium permeant channels, such as the Transient receptor potential channels (TRPC), were linked to changes in the Ca^2+^-induced plasticity and a loss of working memory [36,41]. In the *Drosophila* memory mutants, *dunce* and *rutabaga*, the PTP is altered [42], suggesting that post-tetanic plasticity plays an important role in synaptic plasticity associated with short-term memory.

### 3.2. Short-Term Memory Alterations Are Associated with Dlg Loss of Function

We observed that the *dlgS97* mutants exhibited strong defects in aversive and appetitive short-term memory paradigms, showing that this protein is necessary for the olfactory associative learning, independent of the conditioning paradigm applied. Considering the synaptic perturbations exhibited by the *dlg* mutants in the larval NMJ [13,15] and the post-tetanic changes that we observe, these learning defects are likely to arise from alterations in the synaptic function and plasticity. The evidence that this mutant learning phenotype can be rescued by the expression of DlgS97 in the mushroom bodies reveals that it is in this structure that this protein is necessary for olfactory learning. The rescue experiment was performed using the OK107 driver, which is a driver that is strongly expressed in all of the Kenyon cells, however, as occurs in the other drivers used to express in the MB, OK107 is also expressed but not as strongly as in the other structures, including the optic lobe and some cells in the antennal lobe [43]. Therefore, we cannot discard that other structures or cells included in the expression of the OK107 driver participate in the memory process.

Both of the types of associative paradigms, aversive and appetitive, depend on the mushroom bodies, where the subgroups of the Kenyon cells are responsible for associating specific odors to the stimulus [44,45]. The mushroom bodies also receive afferents from the dopaminergic and octopaminergic neurons, which participate in defining if the response is aversive or attractive [46,47]. These results, together with the rescue experiments, support a main role of the MB in the Dlg-dependent memory defects.

*Drosophila*’s central brain has approximately 100,000 neurons, of which more than 2000 are Kenyon cells [48,49], which correspond to the mushroom body, where the decoding of the sensory information of the odors processed in the antennal lobe occurs. The mushroom bodies correspond to the structure where the greatest expression of DlgS97 occurs in adults. Taking this into account, the molecular structure and the integrating role of this structure, may be the main characteristics that make it more susceptible to the decrease in the expression of the DlgS97 protein [30,50,51]. We determined that a decrease in the DlgS97 expression in the Kenyon cells, using the 117Y-GAL4 driver, was associated with a strong decrease in the learning index [43,52]. Dlg is present both pre- and post-synaptically in the NMJ, and the expression in the Kenyon cells is strong in all of the projections [14]. Since the mushroom bodies are postsynaptic for the sensory input and presynaptic for the performance of the behavior, it is not possible at this point to confirm that Dlg is needed pre-synaptically or post-synaptically for short-term memory. The observation that the *dlgS97* expression is not required in the Kenyon cells located in the α’/β’ lobe may suggest that this protein is not required in this subset of cells, or that impairment of aversive learning is only attained when the *dlgS97* downregulation is performed in all of the subsets of the Kenyon cells. On the other hand, the observation that the *dlgS97* downregulation in the ascending olfactory learning does not impair the olfactory learning supports the lack of mutant defects in the evoked response in physiological calcium observed in the NMJ synapse, as well as several studies indicating that the genetic or pharmacological disruption of the mushroom bodies prevents aversive associative learning, without grossly disturbing the sensorimotor responses to the stimulus [53,54,55,56,57].

### 3.3. Dlg-NMDARs Association

The hypomorphic mutant for the *Drosophila* NMDAR gene (dNR) exhibits defects in the olfactory learning [50] and is co-immunoprecipitated with Dlg [30] (Figure 3), supporting that they are in the same complex. Here, we confirmed the association between DlgS97 and dNR by immunoprecipitation [30,50]. Additionally, our in vivo association experiments show that the tail of dNR2 depends on DlgS97 to reach the synapse, the main protein present in the adult brain. Thus, it is plausible that DlgS97 is acting post-synaptically, as in vertebrates’ CNS, to stabilize the dNRs [15,30,52].

These results suggest that the memory defects displayed by the *dlgS97* mutants are associated with its interaction with dNRs. However, do the short-term plasticity defects observed in the larval NMJ provide a mechanism for the short-term memory defects observed in the adults? We believe that they do only partially; the mechanisms observed in the NMJ are of presynaptic origin and do not involve the dNR, while the short-term memory could be linked to the Dlg–dNR association. We cannot discard a presynaptic component for the memory defect observed, considering that the mushroom bodies are presynaptic for other structures, such as the central complex where it is thought that the memory could be stored. The short-term plasticity defects in the larval NMJ might be part of the mechanism by which the *dlg* mutants display short-term memory deficiency in the adult fly, however, these two different mechanisms might also be independent. For example, an NMDA component of presynaptic origin has been observed in the NMJ through the blockage of glutamate Kainate-type receptor. dKaiR1D, which forms homomers, located in the presynaptic compartment, and is necessary for synaptic homeostasis [58,59]. Similar to NMDARs, dKaiR1D is calcium permeable and is blocked by magnesium [58]. In turn, the Dlg could be acting through only the presynaptic mechanisms in the mushroom bodies, and its association to dNRs might be important only for long-term memory, as was suggested by Wu et al. [31]. In this publication, the immunoreactivity to NRs was particularly concentrated in the central complex and the downregulation of NRs in this structure only affected the long-term memory.

In summary, we show here that the *dlg* mutants display short-term memory defects that we suggest are associated with its direct binding to the dNRs. Additionally, based on the presynaptic defects in the short-term plasticity observed in the NMJ, a presynaptic component, mainly due to an abnormal calcium accumulation in the terminal, could be playing a role. However, we cannot discard that the *dlg* mutants’ deficiency in larvae and adults is associated with two different mechanisms, since each system has its peculiarities.

## 4. Materials and Methods

### 4.1. Drosophila Stocks and Genetics

The following strains were used: *y*,*w* and CS were used as the control; *y*, *w*, *dlg^XI-2^*; *y*, *w*, *dlgS97^5^*; *y*, *w*, *dlgA^40.2^*; (Bloomington Stock Center, Bloomington, IN, USA). All of the strains were backcrossed six-fold with Canton-S to standardize the genetic background (Cantonization). For the adult associative olfactory learning experiments, the strains used were: W(CS); OK107-Gal4 (kind gift from Dr. Jorge Campusano); UAS-RNAi-*dlg* (Vienna *Drosophila* Resource Center, AT); UAS-dsRNA-*dlg1*; Orco-GAL4; GH146-GAL4; Tenm-GAL4;117y-GAL4; C305A-GAL4; Elav-GAL4 (Bloomington Stock Center, Bloomington, IN); Ok107-GAL4; UAS-DlgS97 (Kind gift from Dr. U. Thomas). For the dNR1 and dNR2 overexpression, UAS-GFP-CD8-dNR1 and UAS-GFP-CD8-dNR2 flies were used. All of the flies were grown at 22 °C.

### 4.2. Generation of NMDAR CD8 Tails and Flies

The sequence of the last 144 amino acids of *Drosophila* NMDA-R1 and NMDA-R2 were synthesized by Genewiz (South Plainfield, NJ, USA) and cloned in the pUAST-CD8-GFP vector, using SacII and Xba restriction site, introduced in the synthesized region during synthesis to obtain pUAST-CD8-NR1-GFP or pUAST-CD8-NR2-GFP. The DNA was then sent to the GenetiVision injection service (Houston, TX, USA) to obtain the transgenic flies.

### 4.3. Electrophysiology

The postsynaptic currents from specific genotypes were recorded at segment A3 of the ventral longitudinal muscle six, as indicated in the third instar larvae using a two-electrode voltage clamp with a −80 mV holding potential, as was previously described [13,60]. The motor nerves innervating the musculature were severed and placed into a suction electrode, so the action potential stimulation could be applied at the indicated frequencies, using a programmable stimulator (Master8, AMPI; Jerusalem, Israel). The pre-synaptic excitation was induced with a square wave of 10 V of amplitude and 100 ms. of duration. The modified HL3 solution in mM: 10 NaHCO_3_, 5 KCl, 4 MgCl, 5 HEPES, 70 NaCl, 5 Trehalose, 115 Sucrose, pH 7.2. The final Ca^2+^ concentration was adjusted to 2 mM.

### 4.4. Electrophysiological Data Acquisition and Analysis

Electrophysiological data acquisition and analysis were performed using the National Instrument DAQ system, controlled by the WinEDR V3.8.2 software (University of Strathclyde, Glasgow, UK). The analyses were performed using Clampfit-10 software (Axon pClamp-10.7, Molecular Devices, Sunnyvale, CA, USA) and Origin-2017 (OriginLab, Northampton, MA, USA). The kinetics analyses were performed as previously described [13,21].

### 4.5. Adult Learning and Short-Term Memory Paradigm

The 3 to 5 days old flies were trained using an aversive olfactory conditioning, in which the odorants 3-octanol (OCT) or 4-methyl cyclohexanol (MCH) were used as a conditioned stimulus (CS), and a 90 V pulsed (12 × 1.5 s) electric shock as an unconditioned stimulus (US) [61]. For each assay, an average of 40 adult flies was used, with no sex distinction, except in the rescue experiments, in which only the male flies were used, since only males carried the mutant gene and the GAL4-UAS combination after crossing the female *dlg* (X chromosome) mutants carrying the Gal4 to the males carrying the UAS-rescue construct. A T-maze apparatus was used for the learning and memory experiments [5]. The flies were exposed for 1 min to one of the odors paired with electric shock reinforcement (CS+) and, 30 s later, the second odor was presented for another minute, without the electric shock (CS-. A total of 2 min after conditioning, the flies were transferred to the T choice point, where each odor was presented simultaneously in different test tubes. The flies had 2 min to choose between the odors. The learning index (LI) was calculated as the number of flies that choose the CS-odor minus the number that preferred the CS+, divided by the total number of flies. The final LI considers the average of the learning index, using MCH or OCT as a CS+ in different sets of flies, so, for each strain, both of the odorants were used as CS+ alternately, in separate assays. In this determination, for every strain evaluated, we performed 10 assays, obtaining 5 LI for each. In the appetitive olfactory conditioning, the same methodology was applied, changing both the motivational state of the fly through food deprivation (16–20 h) and the use of an aversive conditioned stimulus to an appetitive one, using filter papers soaked with sucrose solution [62]. The LI, in this case, is calculated as the number of flies that choose the CS+ odor minus the number that preferred the CS–, divided by the total number of flies. The final LI, as before, was the average of the learning index, using MCH or OCT as a CS+ in the different sets of flies. All of the assays were performed in a dark room at 25 °C degrees and 60% humidity. Before the behavior assays, odor acuity and a concentration of the odors (1:1000) that did not evoke a basal preference were determined. For these experiments, the flies of the UAS or Gal4 strains were raised in a 25 °C degree environment.

### 4.6. Immunoprecipitation Assays

A total of 25 μL of protein G or protein A agarose beads (per assay) were washed two times with 1 mL of TBST buffer (150 M NaCl, 20 M Tris HCl, 0.02% Tween 20). Centrifuging took place for 5 min at 3000 rpm each time, then the supernatant was removed and 100 μL of TBST together with 3 μL of antibody (anti-Dlg-PDZ-4F3 (DSHB) and polyclonal anti-NR2 84S 1:5000 [31]) were added; then, it was left to incubate the mixture at 4 °C for 2–3 h in rotation.

For the preparation of the sample, 2–3 mL of adult flies were taken (per test) previously frozen in liquid nitrogen. The flies were passed through a sieve that allowed for the recovery of the heads, that were subsequently homogenized on ice with 1 mL of RIPA buffer (150 mM NaCl, 20 mM Tris HCl pH 7.5, 1 mM EDTA, Na+ deoxycholate 0.5%, NP40 0.5%) with protease inhibitors. The resulting homogenate was centrifuged at 3000× *g* for 5 min, the supernatant was taken and pre-cleared for 1–2 h with 25 μL of protein A or G agarose beads. The samples were subjected to SDS-PAGE and the gel was stained with silver or zinc as described below.

### 4.7. Immunostaining

The immunofluorescence (IF) of the adult brains was performed using the previously described protocols [63]. Anti-GFP (Invitrogen), Bruchpilot (nc82, DSHB), Futsch (22C10, DSHB), Repo (8D12, DSHB), and Elav (Elav-9F8A9, DSHB) antibodies were employed. The secondary antibodies were purchased from Jackson Laboratories and used 1:200. The larval dissections were performed and the NMJ immunofluorescence of the larvae was performed as in Astorga et al., 2016. The Cy-5-conjugated anti-HRP and HRP- or fluorescent-coupled secondary antibodies (1:200; Jackson ImmunoResearch, West Grove, PA, USA) and mouse anti-GFP 3E6 (1:200, Invitrogen, Waltham, MA, USA) were employed.

### 4.8. Image Acquisition and Analysis

The IF images were acquired with a confocal microscope (Olympus FV1000), deconvolved, and quantified using ImageJ (U.S. National Institutes of Health). The data were collected in Excel (Microsoft, Redmond, WA, USA) to be processed. For the fluorescence intensity assays, the relative fluorescence corresponds to the intensity value of the GFP signal tightly close to the HRP labeling area (peri-synaptic) compared to the postsynaptic enlarged area of 3 μm around the HRP mark, which recapitulates the Sub synaptic reticulum area (SSR); this fluorescence ratio corresponds to the peri-synaptic/SSR quantification). In complement, the fluorescence ratio, between the SSR and the different muscle GFP signals away from the synapse area (Muscle ROIs), corresponds to (SSR/Muscle). The number of data correspond to the number of larvae or adult images.

### 4.9. Calcium Reporter (GCamp) Experiments

For the GCaMP imaging experiments, the wandering third instars larvae expressing Gcamp6f on the muscle (C57 > Gcamp6f) were dissected in the HL3 without calcium and magnesium, after washing three times, the filet was incubated for 1 min in HL3 with 0.5 mM CaCl_2_ and A2-NMJs with type 1b boutons in the muscles 6/7 and were imaged in HL3 without magnesium (in mM): 70 NaCl, 5 KCl, 0.5 CaCl_2_, 0 MgCl_2_, 10 NaHCO_3_, 5 trehalose, 123 sucrose, and 5 HEPES, pH 7.2.

The images were acquired in an Olympus (Center Valley, PA, USA) BX61WI coupled to a CCD camera Hamamatsu ORCA-R2 controlled by Cell^R software (Olympus, Tokyo, Japan), with an objective lens 60X LUM PlanFl water immersion with a numerical aperture of 0.9. After acquiring a 10–20 s base line at 10 Hz, NMDA 1 mM or a KCl 100 mM puff of 300 ms was delivered by picospritzer Narishige (Tokyo, Japan), using the same pipette for the control-paired batch experiments. For the MK-801 experiments, the filet was pre-incubated for 2 min in 2 mM of the inhibitor before the stimulation with NMDA or KCl. The recorded images were processed using the following formula: (∆F/F0 = F − Fbkg/F0 − Fbkg), where F is the fluorescence intensity measured in type Ib boutons of muscle 6/7; Fbkg is the background signal; and F0 is the intensity at the initial time. The images were analyzed using ImageJ 2.0.0–rc–54/1.51f software (NIH, Bethesda, MD, USA).

### 4.10. Data Analysis

For the confocal image assays, and learning assays, the data obtained from the images were tabulated in Excel (Microsoft) and subsequently analyzed in GraphPad 8 Prism, each dataset corresponding to the average quantification of respective larvae/adults. Mann-Whitney (non-parametric) was used for the control analysis versus RNAi or mutant condition and Kruskal–Wallis non-parametric analysis, with Dunn post-hoc test to compare the control versus different RNAi/mutant conditions.

## Figures and Tables

**Figure 1 ijms-23-09187-f001:**
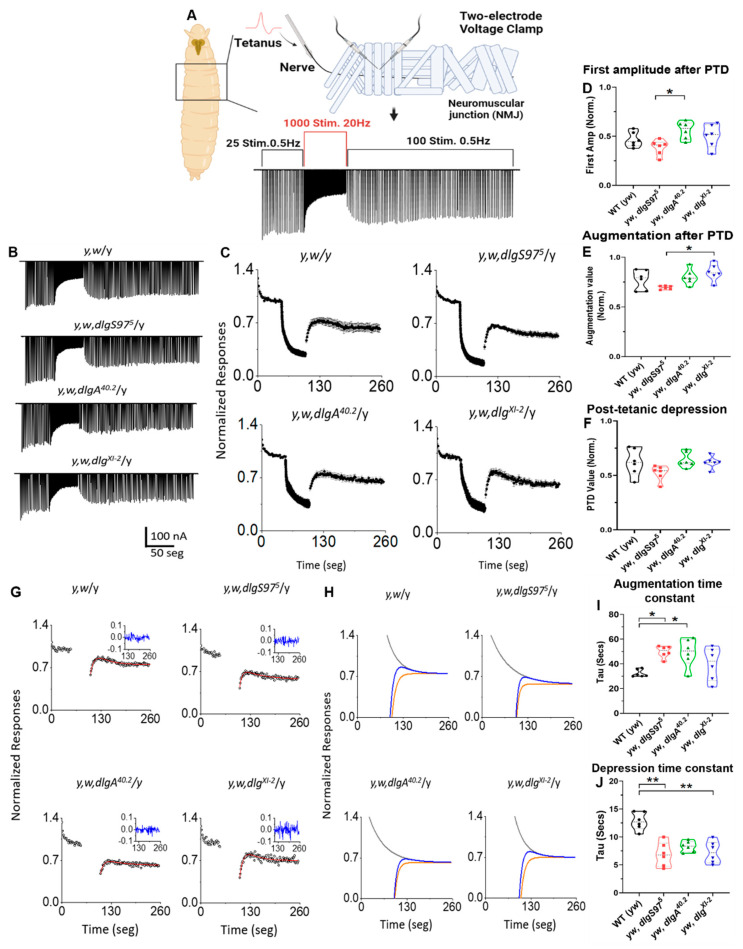
*dlg* mutants can sustain post-synaptic responses after HFS, alter augmentation, and depression time-course. (**A**,**B**) Nerve-evoked junctional currents in control and *dlg* mutants elicited during a paradigm of stimulation to induce post-tetanic potentiation (Created with BioRender.com). The conditioning episodes are the 50 s at 20 Hz and 0.5 Hz for test stimulation. The recordings were obtained at 2 mM of external Ca^2+^; (**C**) Average responses normalized to the mean pre-tetanic value in the control and *dlg* mutants; (**D**) Average responses of the first normalized amplitude after the tetanic stimulation; (**E**) Augmentation of post-tetanic responses, corresponding to the maximum value reached after the conditioned tetanus; (**F**) Post-tetanic depression, corresponding to the average of the 10 last responses. Bars indicate the standard error, *n* = 6; (**G**) Normalized nerve evoked responses from representative recordings during the test stimulation at 0.5 Hz before and after the high frequency of nerve stimulation episodes in control and *dlg* mutants. The inset shows the residual plot of the fitted model to the data, the line in red is the fitted curve; (**H**) Individual post-tetanic augmentation (grey) and depression (orange) with convolved function (blue) in each genotype; (**I**) The time constant for augmentation (Kruskal–Wallis, Dunn test * *p* < 0.05); (**J**) The time constant for depression (Kruskal–Wallis, Dunn test ** *p* < 0.01). Bars indicate the standard error, *n* = 6.

**Figure 2 ijms-23-09187-f002:**
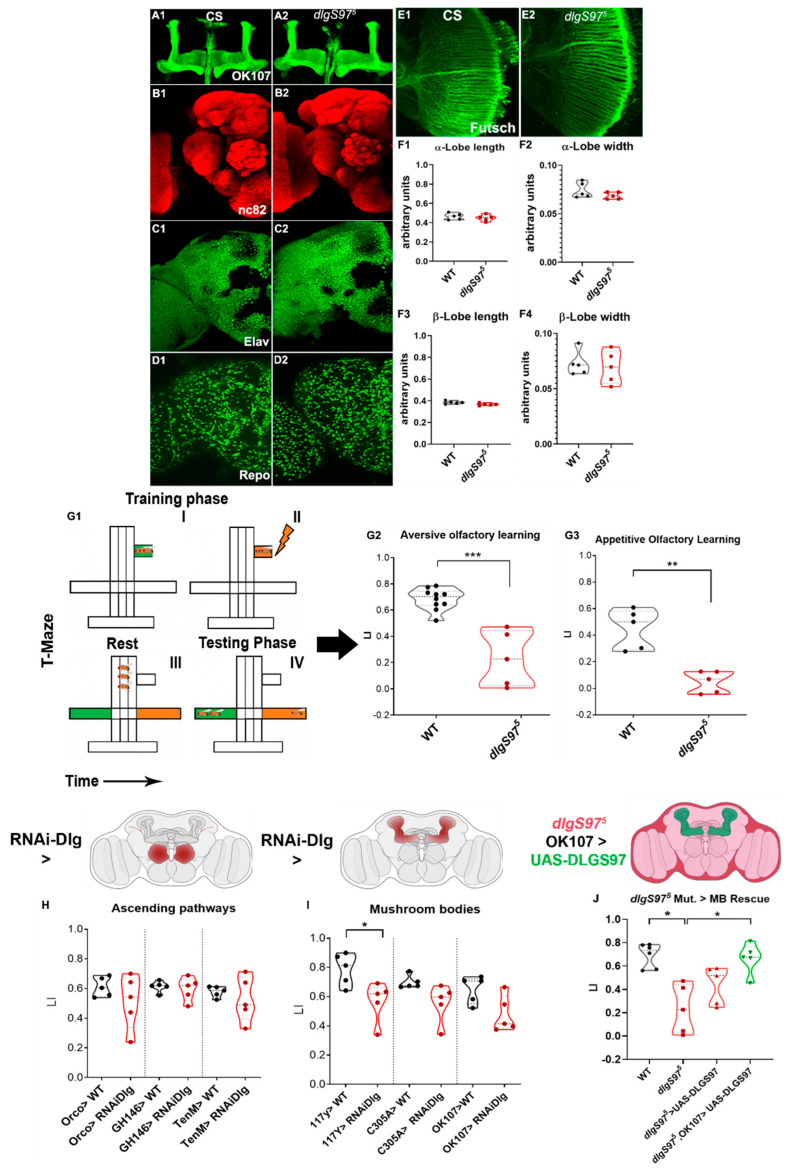
DlgS97 is necessary in the mushroom bodies, but not required in the ascending olfactory pathway for associative olfactory learning. (**A1**–**F4**) Expression of CNS markers in *dlgS97^5^* mutants in adult brains; Immunofluorescence of adult control (**A1**–**E1**) and *dlgS97^5^* (**A2**–**E2**) mutant brains; (**A**) brains expressing CD8–GFP with the OK107 promoter expressing in all Kenyon cells to study the structure of the MB neuropile; (**B**) brains labeled with an antibody against the active zone protein bruchpilot (nc82 antibody) to observe the general structure of the brain; (**C**) Brains labeled with the antibodies against the neuronal-specific protein Elav; (**D**) Brains labeled with the antibodies against the glial-specific protein Repo (**E**) optic lobes labeled with the antibody Futsch against the cytoskeleton protein MAP to observe the axonal connectivity. Mutant brains show mushroom bodies (**A**) antennal lobe and optic lobe normal structure as well as the general neuronal and glial density; (**F1**–**F4**) Measurements of different parameters of the MB in control and mutant brains. No significant differences were found; (**G1**) Cartoon showing the protocol to induce short-term avoidance memory in a T-maze; The green color represents the odor not paired with an aversive/appetitive stimulus, while the orange color represents the odor paired with these unconditioned stimuli; flies are exposed to an CS-odor (I), then with the second odor and the electric shock (II) after one minute rest (III) the flies are offer to choose between the two odors (IV) (**G2**) Learning indexes from aversive olfactory conditioning in control and adult *dlgS97* mutants (*N*: 5–10), Mann–Whitney, *** *p* < 0.0001; (**G3**) Appetitive olfactory conditioning learning index in adult control and dlgS97^5^ mutants (*N* = 5), Mann–Whitney, ** *p* < 0.01. Each value represents the mean and standard deviation of the mean; (**H**) Learning index of control and *dlg*-KD in ascending pathway neurons (Orco-GAL4/Gh146-GAL4/TenM-GAL4>RNAi-*dlg*) (*N*: 5). Cartoon above illustrates the olfactory ascending pathways; (**I**) Learning index of control and *dlg*-KD In mushroom bodies, projection neurons and interneurons (117y-GAL4/C–305a–GAL4/OK107-GAL4>RNAi-*dlg*) adult flies; Mann–Whitney, * *p* < 0.05 (*N* = 5). The cartoon above illustrates the Mushroom Bodies (**J**) Learning index from control, *dlgS97^5^* mutant, *dlgS97^5^* mutant UAS control (*dlgS97^5^*, UAS–DlgS97), and mushroom bodies DlgS97 rescue (*dlgS97^5^*, UAS–DlgS97; OK107-GAL4) strains. The cartoon above illustrates the DlgS97 rescue in the *dlgS97^5^* mutant. One-way ANOVA (Kruskal–Wallis, Dunn: * *p* < 0.05; *N* = 5) Brain models created with BioRender.com.

**Figure 3 ijms-23-09187-f003:**
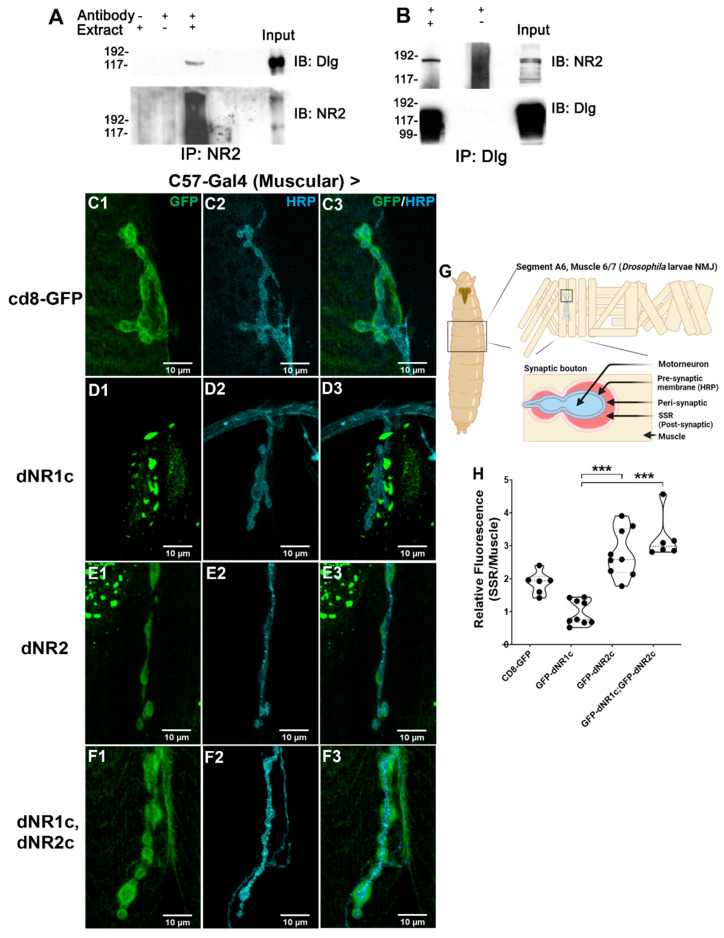
Dlg interacts with Glutamatergic NMDA Receptors. (**A**,**B**) Immunoprecipitation experiments used antibodies against NR2 subunit (**A**) or Dlg (**B**) to precipitate the complex, in turn, NR2 (**B**) and Dlg (**A**) antibodies were used to determine the interactions; (**C1**–**F3**) Representative images of the distribution of dNR1 (dNR1c) and dNR2 (dNR2c) tails overexpressed in the muscle in control flies detected by immunofluorescence against GFP (**C**–**F**) to label the constructs CD8–GFP-dNR1 (dNR1c) or CD8-GFP-dNR2 (dNR2c), antibodies against HRP were used to label the presynaptic boutons (**C2**–**F2**); the merged images are in (**C1**–**F3**); (**C1**–**C3**) expression of CD8-GFP as control of the distribution of a membrane protein. (**D1**–**D3**) distribution of NR1 tail; (**E1**–**E3**) distribution of NR2 tail; (**F1**–**F3**) distribution of the GFP label when dNR1c and dNR2c are co-expressed; (**G**) *Drosophila* larva NMJ synaptic buttons cartoon representing the HRP staining which labels the neuronal membrane. Peri-synapse represents the synaptic area proximal to the neuronal membrane, while the SSR represents the postsynaptic specialization that surrounds the synaptic button. (Created with BioRender.com); (**H**) Quantification of the ratio of GFP intensity between the SSR area and a random region in the muscle in the different genotypes. Kruskal–Wallis, Dunn: *** *p* < 0.001; *N* = 6–9 boutons/2–3 larvae.

**Figure 4 ijms-23-09187-f004:**
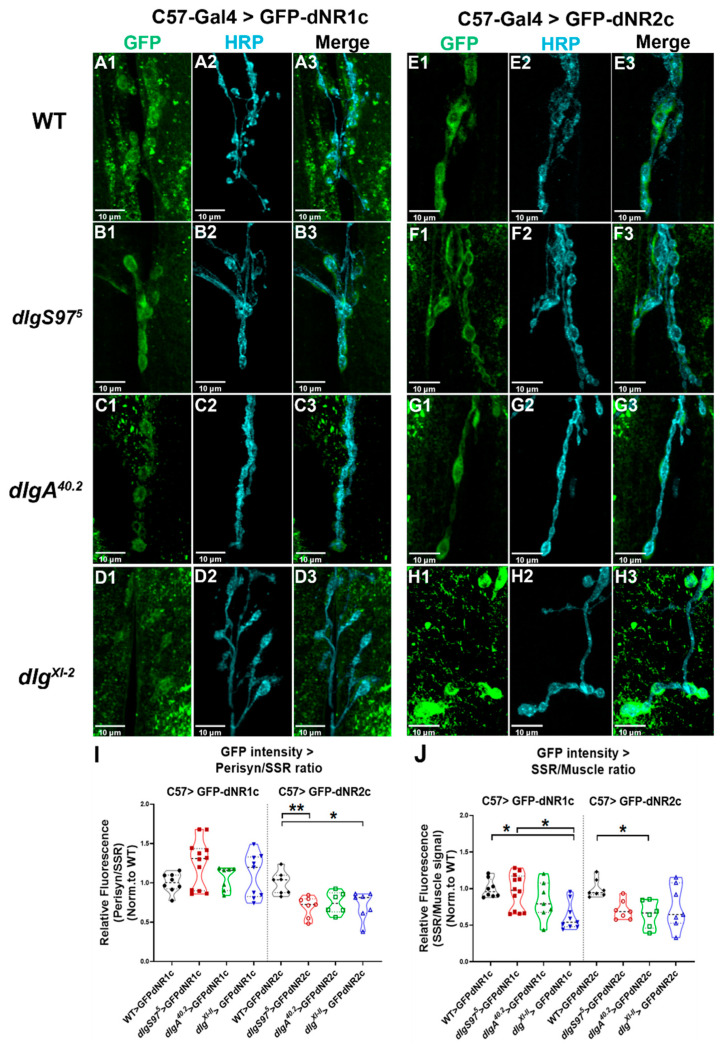
dNR1and dNR2 tails depend on Dlg to reach the synapse. (**A1**–**H3**) Representative images of the distribution of dNR1 (dNR1c) and dNR2 (dNR2c) tails overexpressed in the muscle in control (**A**,**E**) or *dlg* mutant flies (**B**–**D**,**F**–**H**) detected by immunofluorescence against GFP (**A**,**H**) to label the constructs CD8-GFP-dNR1 (**A**–**D**) or CD8-GFP-dNR2c (**E**,**H**) to observe its distribution in the muscle. Antibodies against HRP were used to label the presynaptic boutons (**A2**–**H2**), merge images are shown (**A3**–**H3**); (**I**) Quantification of the fluorescence intensity in the synapse normalized by the fluorescence in the cytosol for all the genotypes. GFP intensity Peri-synaptic/Sub-synaptic reticulum (SSR) ratio (GFP signal tightly close to HRP labeling area compared to the postsynaptic enlarged area around HRP mark). C57-Gal4 > GFP-dNR2c = Kruskal–Wallis, Dunn: ** *p* < 0.01; *N* = 7 boutons—2 larvae; *p* * < 0.05 N = 7 boutons-2 larvae (**J**) GFP intensity SSR/Muscle ratio (GFP signal restricted to the postsynaptic enlarged area around HRP mark versus Muscle ROIs area). C57-Gal4 > GFP-dNR1c = Kruskal–Wallis, Dunn: * *p* < 0.05; *N* = 8/12 boutons—2/3 larvae. C57-Gal4 > GFP–dNR2c = Kruskal–Wallis, Dunn: * *p* < 0.05; *N* = 7 boutons—2 larvae.

## Data Availability

Not applicable.

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
