# Peer review of "Dlg Is Required for Short-Term Memory and Interacts with NMDAR in the Drosophila Brain"

_ijms, 2022, doi:10.3390/ijms23169187_

Round 1

Reviewer 1 Report

In this study, authors explore the role of the scaffolding protein Dlg in Drosophila. Authors show that Dlg mutant larvae display changes in the time-course of a presynaptic-related form of short-term plasticity in the NMJ. Authors then tested the requirement of Dlg for associative olfactory learning in adult flies and provide strong evidence that Dlg in the mushroom bodies is involved in this form of learning. Authors finally suggest that Dlg interacts with NMDARs and that it stabilizes the receptors at synapses. Whilst this study describes an important, yet not novel, role of Dlg proteins in short-term synaptic plasticity and memory, the link between the different studied aspects (post-tetanic plasticity in larvae NMJ; olfactory learning in adult; Dlg-NMDARs interaction) is rather weak. My main concern is that some of the interpretations made by the authors are not supported by the experiments performed, please see my major comments.  

Major comments:

-        Authors suggest that “short-term memory defects are a consequence of the disruption of the NMDAR localization or abundance” (lines 103-104). Authors propose a link between deficits in olfactory learning and Dlg-NMDAR interaction. This suggestion is only based on the interaction of Dlg with NMDAR by immunoprecipitation in adult brains (which in itself is not convincing, please see comment below). Proof of interaction does not demonstrate the involvement of Dlg in the synaptic localization of NMDARs, and does not show that this mechanism supports olfactory learning. In this regard, authors should at least show that the synaptic abundance of NMDARs in adult brain mushroom body synapses is altered in Dlg mutant flies. If possible, authors should also attempt at rescuing the defect in learning index of the Dlg mutants by expressing NMDARs in MB. Have the olfactory learning paradigms assessed in this manuscript been shown to dependent on NMDARs? If so, please cite the studies.

-        Another weak point concerns the suggestion by the authors that Dlg has NMDAR-independent roles in the larvae NMJ (“We determined that NMDARs in the NMJ are not present, supporting an additional role for Dlg in short-term memory independent of NMDARs” lines 106-107). The short-term plasticity paradigm tested relies on presynaptic mechanisms. Authors prove in fig S2 that NMDARs are not present in the muscle of larvae NMJ; however, response to local application of NMDA suggests presynaptic localization of the receptor (fig S2). Authors should prove that NMDARs at motoneurons are not required for the time-course of post-tetanic plasticity.

-        The approach used by the authors to demonstrate that Dlg is required for NMDARs to reach the synapse is not really convincing. Authors overexpress the intracellular portion of NMDARs in larvae muscle in control or Dlg mutants and assess synaptic enrichment (fig 3 and 4). However, the control construct CD8-GFP is enriched at NMJ at similar levels than the overexpressed NMDAR tails (fig 3G), showing that synaptic enrichment of membrane bound proteins is not specific to NMDARs. This puts into question the reliability of this approach to study the specific mechanism of Dlg-NMDAR interaction and synaptic abundance of the receptor. Also, it is hard to understand why the authors quantified the relative fluorescence Presyn/Postsyn considering that expression of the constructs is muscle specific, and so, no expression at the presynaptic side is expected.

-        Considering my major concerns, I feel that conclusions drawn in the discussion are overstatements and should be softened (mostly in lines 409-414).

-        The writing of this manuscript is at some points careless and confusing, therefore deserving special attention and rewriting (some examples can be found at: lines 31-34; lines 71-73; lines 112-131; lines 171-184; lines 225-228; lines 232-237, etc).

Minor comments:

-        Please revise the punctuation throughout the text, including legends.

-        Lines 137 and 139: the frequency of stimulation indicated in the text (0.2Hz) is not in conformity with the one indicated in fig1A (0.5Hz). Please correct.

-        Line 140: “average pre-tetanic response (dashed line in Figure1C)” not visible in the figure, please indicate.

-        Lines 142-143: please cite the studies in which these strains were generated.

-        Lines 145-146: please indicate figure 1D after “The initial value after the tetanus was in folds 0.47 ± 0.077 in control; 0.39 ± 0.077 in dlgS975; 0.56 ± 0.089 in dlgA40.2; 0.50 ± 0.119 in dlgXI-2 showing no differences between mutants and control”.

-        Lines 151-152: please indicate figure 1F after “The final value was in folds 0.63 ± 0.11, in Control; 0.54 ± 0.07, in dlgS975; 0.66 ± 0.07 in dlgA40.2 0.64 ± 0.06 in dlgXI-2”.

-        Lines 161-162: The following does not correspond to figure 1D and needs to be removed “and average of the last ten amplitudes of the 0.5 Hz of test stimulation.”

-        Lines 167-168: How were graphs in fig 1H obtained? How do graphs in 1H differ from graphs in 1G? Please provide a better description of the data presented in 1H and how it was generated. How do the curves in orange and blue compare to the fitted curve (red) in 1C? Why is the starting point of the grey curve not the same in all genotypes? It is difficult to reconcile why the y axis is the same between 1G and 1H but the curves start at different y-values. The main text provides a description of the data analysis performed (lines 171-184), but it is confusing and should be rephrased for better comprehension. Also, please make uniform the terms used in the main text and in the legend: orange line in 1H is potentiation or augmentation?

-        Line 175: define A and D.

-        Line 183: There are no insets in fig 1H, please remove “(the insets show the residual plot for each fit)”.

-        Lines 195-197: Authors seem to imply that the short-term plasticity dependent on DlgS97 and described in ref 13 underlies the paradigm of short-term memory tested on figure 2. Please include references supporting this.

-        Legend figure 2: A cartoon is provided before 2H-J that is not referred in the legend.  Please provide a brief explanation in the legend of this cartoon.

-        Line 212: “Neuronal and glial density” was not determined. Should be replaced by “gross changes in the distribution of neurons and glia”.

-        Lines 214-215: Please define orange and green in the legend.

-        Line 236: What do the authors mean with “showing that despite the pathway of the short-term memory involved”?

-        Lines 238-240: please include references supporting that “associative learning has been shown to be highly dependent on the function of the Kenyon neurons”.

-        Line 241: Please describe the effect of Dlg1-RNAi construct on the expression of DlgS97 and DlgA. This downregulation strategy affects DlgS97, DlgA or both?

-        Lines 247-255: Authors should comment on the lack of significant effect when the OK107-GAL4 driver was used (in comparison to 117y-GAL4 driver) (figure 2I) and the inability of UAS-DlgS97 to rescue the phenotype.

-        Lines 253-255: In figure 2J, rescue is accomplished with OK107 driver (which as authors indicate “enables a broader expression, including Kenyon cells, projection neurons and interneurons of the antennal lobe”). Moreover, downregulation of Dlg in all Kenyon cells (117y driver) did not fully recapitulate the effect of the mutant dlgS975 (fig 2G) or pan-neuronal downregulation of DlgS97 (fig S1) on the learning index. Together, this suggests that Kenyon cells are not the only cells in which DlgS97 plays a role in olfactory learning. Please rephrase.

-        Fig 3: On panels A and B, indicate molecular weight on WBs. In B, the negative control without extract shows a strong signal in the immunoblot against NR2 (IP:Dlg). This observation suggests lack of specificity of the NR2 antibody and puts into question the conclusions made. The authors should provide a cleaner result for this IP. Moreover, an immunoprecipitation assay does not offer proof of direct binding, and so, the word “directly” should be removed from the title of the legend.

-        Fig S2: Material & methods is missing for the experiments contained in this figure. Please define VNC in the legend. For 3E, 2F and 2G, please provide the plot of intensity of peaks (as in 2C).

-        Lines 278-281: It should be indicated along the text that expression of the tails of NMDARs in the muscle was accomplished by the C57 muscle-specific driver.

-        Lines 300-304: Description of the results does not agree with the quantitative data provided in 4J-K. (e.g. dlgA40.2 mutants show altered synaptic/SR fluorescence suggesting reduced postsynaptic localization of NR2; synaptic/SR relative fluorescence does not significantly differ between DlgS97 and controls for NR1 and NR2). Considering that NMDAR constructs are expressed specifically in the muscle, the quantification of the Presyn/Postsyn relative fluorescence does not seem relevant. Please comment.

-        Line 318: indicate panel (K) before “GFP intensity Synaptic / SR ratio”.

-        Lines 380-383: Please include reference for the described observations.

-        Line 472: Please provide an explanation for only using males in the rescue experiment.

-        Lines 473-475: The order described here does not correspond to the order represented in fig2 G1, please make it consistent.

-        Lines 479-482: If I understood correctly, the same group of flies was assayed twice in the learning paradigm with the odorants OCT and MCH as CS+. In case yes, authors should indicate the time between the two assays.

-        Line 490: Please provide information of the antibodies used for the IP.

-        Line 512: define IF.

-        Line 516: What is the size of the “enlarged area around HRP mark”? Does it encompass the postsynaptic region? Authors should provide representative images of pre/post-labeled NMJ showing the HRP ROI, the enlarged ROI around HRP mark (postsynaptic) and the muscle ROIs. How is the muscle ROI defined?

Author Response

We thank the reviewer fr the detailed revision. All the comments were taken in account and we beleive that the manuscript improved thank to them.

Reviewer 2 Report

This is a nice study showing that the scaffold protein Dlg is required for short-term memory in Drosophila, both in dependence on the NMDA receptor and independently. The manuscript is well written and the data are very convincing. I only have several minor comments.

(1) I believe that in the authors' list, the symbol $ has to be changed into & (or vice versa).

(2) Throughout the manuscript, there are several typos ("immuneprecipites") and editing errors (missing spaces, wrong decimal sign. Please check carefully.

(3) The figures should be improved. Most importantly, several titles and legends are simply to small to be read. Also, consistency could be improved. In Fig. 1A, one could indicate where/when the measurements shown in Fig. 1D-F are taken.

(4) I am puzzled by the learning indices in Fig. 2G2/3. Shouldn't aversive and appetitive learning lead to negative vs. positive indices. According to the methods, the same formula was used to calculate the indices.

(5) Is there a special reason why o0nly male flies were used in the rescue experiment?

(6) I guess the "please add" in the Funding section can be removed.

Author Response

This is a nice study showing that the scaffold protein Dlg is required for short-term memory in Drosophila, both in dependence on the NMDA receptor and independently. The manuscript is well written and the data are very convincing. I only have several minor comments.

Thank you very much for the positive comment

(1) I believe that in the authors' list, the symbol $ has to be changed into & (or vice versa).

Thank you for spotting this error, we have corrected it

(2) Throughout the manuscript, there are several typos ("immuneprecipites") and editing errors (missing spaces, wrong decimal sign. Please check carefully.

We did a general revision of the text and hope to have solved all the spelling and writing issues. 

(3) The figures should be improved. Most importantly, several titles and legends are simply to small to be read. Also, consistency could be improved. In Fig. 1A, one could indicate where/when the measurements shown in Fig. 1D-F are taken.

In lines 144-146, we have improved the description in the legend related to Drosophila larvae NMJ. We have also increased the size of the labels in figures.

(4) I am puzzled by the learning indices in Fig. 2G2/3. Shouldn't aversive and appetitive learning lead to negative vs. positive indices. According to the methods, the same formula was used to calculate the indices.

The indexes were obtained in independent experiments, and the formula was used in an inverse way as the aversive conditioning. Thus is CS+ -CS- instead of CS- - CS+. We have clarified this in the text in lines 508-511

(5) Is there a special reason why o0nly male flies were used in the rescue experiment?

It is a genetic reason since dlg gene is in the X chromosome, females need to be kept over balancer in order to be fertile and viable, thus to cross them with males UAS-Dlg fly, in the F1 only the ubalanced males are mutants. We explained this in lines 248-250

(6) I guess the "please add" in the Funding section can be removed.

Yes, we deleted the word.

Reviewer 3 Report

The manuscript presented by Bertin et al. demonstrated that member of Drosophila Dlg family proteins, specifically, one of the Dlg1 transcripts, is required for short-term memory in Drosophila adults. Their claims are supported by the electrophysiology data collected in larvae neuromuscular junctions and genetic-behavioural data collected in adults. They further showed that the Drosophila Dlg1 physically interacts with NMDAR2 in the adult brain by immunoprecipitation, but not in the larvae brain. Finally, they compared the cellular localisation of both NMDAR1 and NMDAR2 in several Dlg1 mutants and concluded that Dlg1 may have different affinity or interaction mechanisms that are able to distinguish the two NMDAR receptors. 

Generally, the manuscript is scientifically convincing and well written. Whilst the molecular mechanism that differentiates how the NMDAR1/NMDAR2 interact with Dlg1 remains elusive, it is well-beyond the scope of this research and could be left for further investigation. There are a few minor typos and gene nomenclature inconsistencies, such as the "Ca+2" in Discussion (line 326, 329, etc.). Also, although Drosophila has only one Dlg orthologue, it is still recommended to use the official gene symbol Dlg1, rather than simply Dlg, in the text. Finally, if possible, the false-colouring palette used in the immunofluorescence images in Figure 3/4 could be further improved, say, use a cyan/yellow or cyan/magenta combination. The current cyan/green combination is slightly difficult to get the details in the merged two-channel images.

Author Response

Generally, the manuscript is scientifically convincing and well written. Whilst the molecular mechanism that differentiates how the NMDAR1/NMDAR2 interact with Dlg1 remains elusive, it is well-beyond the scope of this research and could be left for further investigation. There are a few minor typos and gene nomenclature inconsistencies, such as the "Ca+2" in Discussion (line 326, 329, etc.). 

We would like to thank the positive comments, we have revised the whole manuscript for typos and grammar errors and corrected them including inconsistencies in the nomenclature. 

Also, although Drosophila has only one Dlg orthologue, it is still recommended to use the official gene symbol Dlg1, rather than simply Dlg, in the text.

As correctly mention by the reviewer, the name of the gene in Drosophila is Dlg1. However, Dlg is one of the synonyms used by multiple publications. To avoid confusions we now define dlg as an abbreviation of dlg1.

Finally, if possible, the false-colouring palette used in the immunofluorescence images in Figure 3/4 could be further improved, say, use a cyan/yellow or cyan/magenta combination. The current cyan/green combination is slightly difficult to get the details in the merged two-channel images.

We used cyan over magenta in order to facilitate the distinction with the muscular dark background, this made it easier to observe the presynaptic component of the NMJ in our opinion. We could change the colors, although it will take time since all pictures would need to be changed for each color channel. Thus we will do it for the final figures.

Round 2

Reviewer 1 Report

I would like to thank the authors for their helpful and convincing answers and their changes in the manuscript.

Please find below just few minor issues:

- Concerning the argument about localization of NMDARs at the NMJ presynaptic terminal. For complete understanding of the interpretation of the results made by the authors, it would be helpful to indicate that removal of the VNC does not disrupt the synaptic structure of the NMJ (pre and postsynaptic terminals still intact). Moreover, it is also necessary to acknowledge that the local application of NMDA with the picospitzer in the NMJ activates NMDARs at the cell bodies level. Only with these two pieces of information, will the reader be able to follow the authors conclusion that NMDARs are not present in the NMJ presynaptic terminal.

- Also, please note that I was still unable to find the “GCamp methodology” both in the manuscript v2 or the supplementary document.

- Please define SSR the first time it appears in the manuscript.

Author Response

I would like to thank the authors for their helpful and convincing answers and their changes in the manuscript.

Thank you for the helpful review

Please find below just few minor issues:

  • Concerning the argument about localization of NMDARs at the NMJ presynaptic terminal. For complete understanding of the interpretation of the results made by the authors, it would be helpful to indicate that removal of the VNC does not disrupt the synaptic structure of the NMJ (pre and postsynaptic terminals still intact). Moreover, it is also necessary to acknowledge that the local application of NMDA with the picospitzer in the NMJ activates NMDARs at the cell bodies level. Only with these two pieces of information, will the reader be able to follow the authors conclusion that NMDARs are not present in the NMJ presynaptic terminal.

we added the following paragraph in line 280

It is important to highlight that the remotion of the VNC does not perturb the structure or function of the NMJ. Additionally, the area stimulated is close to the VNC since we use the A2 segment of the larvae, the more cephalic segment. Therefore, these results in our view confirmed the absence of receptors to NMDA in the muscle and suggest a modulatory function of NRs in the motoneurons in the ventral cord.

  • Also, please note that I was still unable to find the “GCamp methodology” both in the manuscript v2 or the supplementary document.

we are very sorry for this mistake, somehow the change was left behind in one of the documents we corrected. Now is in line 581 

  • Please define SSR the first time it appears in the manuscript.

this is now defined in line 291

Also we did a new spell check